# Computational Approaches to Explore Bacterial Toxin Entry into the Host Cell

**DOI:** 10.3390/toxins13070449

**Published:** 2021-06-28

**Authors:** Weria Pezeshkian, Julian C. Shillcock, John H. Ipsen

**Affiliations:** 1Groningen Biomolecular Sciences and Biotechnology Institute and Zernike Institute for Advanced Materials, University of Groningen, 9712 Groningen, The Netherlands; 2Blue Brain Project, Laboratory of Molecular and Chemical Biology of Neurodegeneration, Ecole Polytechnique Fédérale de Lausanne, 1015 Lausanne, Switzerland; julian.shillcock@epfl.ch; 3MEMPHYS/PhyLife, Department of Physics, Chemistry and Pharmacy (FKF), University of Southern Denmark, Campusvej 55, 5230 Odense, Denmark; ipsen@memphys.sdu.dk

**Keywords:** computational methods, molecular dynamics simulations, coarse-grained simulations, bacterial toxin, membrane remodeling

## Abstract

Many bacteria secrete toxic protein complexes that modify and disrupt essential processes in the infected cell that can lead to cell death. To conduct their action, these toxins often need to cross the cell membrane and reach a specific substrate inside the cell. The investigation of these protein complexes is essential not only for understanding their biological functions but also for the rational design of targeted drug delivery vehicles that must navigate across the cell membrane to deliver their therapeutic payload. Despite the immense advances in experimental techniques, the investigations of the toxin entry mechanism have remained challenging. Computer simulations are robust complementary tools that allow for the exploration of biological processes in exceptional detail. In this review, we first highlight the strength of computational methods, with a special focus on all-atom molecular dynamics, coarse-grained, and mesoscopic models, for exploring different stages of the toxin protein entry mechanism. We then summarize recent developments that are significantly advancing our understanding, notably of the glycolipid–lectin (GL-Lect) endocytosis of bacterial Shiga and cholera toxins. The methods discussed here are also applicable to the design of membrane-penetrating nanoparticles and the study of the phenomenon of protein phase separation at the surface of the membrane. Finally, we discuss other likely routes for future development.

## 1. Introduction 

An effective strategy for many bacterial pathogens to damage a target cell is the secretion of toxin proteins that can modify and disrupt essential cellular processes and lead to cell death. The investigations of these protein complexes and their mechanism of action have contributed significantly to advances in several fields of life science. For instance, important mechanisms in cell biology have been discovered by unraveling specific interactions of toxin proteins with their target cells that are also essential for biomedical applications. To perform their action, bacterial toxins often need to cross the cell membrane and reach a specific location inside the cell, which suggests their great potential for biotechnological and therapeutic applications such as targeting specific tissue, organelle, or toxin-inspired targeted drug delivery vehicles’ design [1]. Additionally, the navigating capacity of bacterial toxins through different barriers of the cell regions point to fascinating mechanisms that are of great interest for biophysicists and chemists. 

A diverse range of experimental techniques have been used to obtain high-resolution atomistic structures, the identification of their cellular receptor and binding affinities and entry pathways, e.g., cryo-electron microscopy and fluorescence microscopy imaging techniques, and biochemical assays [2,3,4,5]. Considerable insight has been gained from such complementary experimental approaches, to the extent that some toxins are now used as tools in cell biological experiments [6]. For instance, cholera toxin was initially used to “identify” raft domains in cell and model membranes [7], and it is now clear that these toxins themselves reorganize raft lipid [8,9]. Because of their importance, more insight into the mechanism of action of bacterial toxins is needed. 

However, for several reasons, experiments alone are insufficient to fully map out toxin entry to cells, and computational modeling is increasingly important: (i) all experimental measurements are prone to some limitations and inaccuracy that can lead to the misinterpretation of the physical principles covering the system, e.g., many experimental investigations use labeled biomolecules, which can perturb the system significantly, leading to incorrect conclusions. Here, it is important to check the proposed physical and chemical mechanism derived from experiments against idealized theoretical model calculations. (ii) Computational models often reveal molecular or cooperative mechanisms which can be tested experimentally, i.e., the computational methods serve as direct input to the design and analysis of experiments. (iii) The actions of many bacterial toxins are complex with sequential and/or concomitant physical and chemical processes involving vastly different length and time scales. Simple biochemical models fail in describing such systems, while multiscale computational modeling helps in combining the processes. (iv) Structural aspects of some bacterial toxins have been subject to evolutionary convergence, e.g., the bacterial Shiga and cholera toxins have common structural motifs for the membrane binding toxin subunits to induce membrane bending and to drive glycolipid–lectin (GL-Lect) endocytosis, although they have no sequence similarity and use different glycolipid receptors [10,11]. This suggests that generic physical properties of the membrane are exploited and computer simulations are an excellent tool for uncovering them.

In this review, we first survey the computational methods for exploring bacterial toxin entry, with a specific focus on all-atom molecular dynamics simulations, coarse-grained simulations, and mesoscopic models, and then summarize the recent advances that have significantly contributed to our understanding. The methods discussed here are also applicable to the design of membrane-penetrating nanoparticles and the phenomenon of protein phase separation at the surface of the membrane. Finally, we discuss other likely routes for future development.

## 2. Computational Methods

Despite the colossal complexity of biological systems, they are still subject entirely to the laws of physics. Thus, in principle, one can study every biological system by performing ab initio calculations, i.e., quantum calculation, on their constituent particles. In practice, this is impossible. Using the most powerful computers and the best available numerical algorithm, the ab initio calculation of systems containing thousands of particles would take decades, yet most biological systems are composed of a much larger number of atoms. Nevertheless, to describe most biological processes, highly accurate computational methods can be constructed without the need for first-principle theories. 

In the past few decades, many powerful computational methods have been developed for the description of biological structures and functions. A large number of biological phenomena are now accessible for computational analysis, ranging from biomolecular dynamics, enzymatic processes, and virus assembly to the structure of small organelles [12,13,14], making computer simulation techniques an indispensable tool. Additionally, they are used for biomolecular engineering and drug design [15,16,17,18], Nevertheless, all computational methods have their strength in specific ranges of time and length scales, while they are inapplicable outside these ranges. Therefore, for many biological processes, several of these methods must be put into play. Multiscale simulation techniques provide such routes of complementary methods where the cooperative behavior of biomolecular assembly can be linked to molecular features. The development of such methods carries a promise of a new level of sophistication in the computational analysis of highly complex cellular phenomena. Besides the application of multiscale simulation techniques in basic research of the cell, it can aid the rational design of new drugs and drug delivery systems in pharmaceutical sciences and accommodate new approaches for combating disease in medical sciences [19,20]. The problem of bacterial toxin entry into the cell is among the first successful application of multiscale simulation methods. For certain classes of bacterial toxins, cell entry is initiated by series of processes: the diffusion of the toxin particle, binding to a membrane receptor, the aggregation of toxin–receptor complexes on the membrane, and large-scale deformation and tubular invagination of the host membrane. Therefore, multiple simulation methods to cover important features of these processes are needed. In the following, we focus on three major computational methods that can well cover the full scale required for bacterial-toxin-related phenomena. These methods are all-atom molecular dynamics simulations, coarse-grained, and mesoscopic models that together can cover a length scale from atomic distance to cellular scale and a time scale up to second (Figure 1).

## 3. All-Atom Molecular Dynamics Simulations

The all-atom molecular dynamics simulation (aaMD) is a unique method to investigate biomolecular systems at atomistic resolutions [14]. After many decades of work on the development of advanced and optimized simulation software packages and accurate force fields, aaMD has become an indispensable tool to capture biomolecular dynamics in the range of 10s of nm and the microsecond timescale [23,24,25,26].

To investigate the mechanism of bacterial toxins’ entry into the host cell and their mechanism of action, aaMD is highly important to explore the dynamics, conformation, and interactions of a single toxin particle and its cellular receptor [27,28,29]. For instance, the dynamics and stability of diphtheria toxin translocation T domain in neutral and low pH and in solution and in the vicinity of anionic membranes has been investigated thoroughly using aaMD [30,31,32,33]. Additionally, the conformation and phase behavior of globotriaosylceramide (Gb3) and monosialotetrahexosylganglioside (GM1), which, respectively, are the cellular receptors of Shiga and cholera toxins, has been explored using aaMD [29,34,35,36]. These studies unraveled several interesting behaviors, suggesting that the length and saturation level of the acyl chain and the composition of the lipid bilayers strongly affects toxic binding and activity. Furthermore, aaMD suggests that CTxB binds efficiently and strongly to lipid bilayers containing GM1 due to the significant amount of direct and water-mediated hydrogen bonds [27,34], supporting the crystal structure data [37].

Another interesting application of MD simulation is to check the reliability of molecular probes that are used for in vitro experiments of toxin behaviors. For instance, Rissanen et al. [34] used aaMD to explore the binding affinity of cholera toxin B-subunit (CTxB) to GM1 and acyl-chain labeled bodipy-GM1 (bdGM1). Interestingly, they found that bdGM1 shows reduced receptor availability in lipid bilayer mixtures, suggesting that the usage of bdGM1 to probe GM1 in cells can lead to an invalid interpretation of experimental data (Figure 2A).

To enter the host cell, many toxins generate their own endocytic pits through cooperative actions initiated by binding to their cellular receptor on the targeted cell membrane [38]. To do so, individual toxic particles need to modify and manipulate the host cell membrane structure. This is characterized by their local membrane curvature imprint, which measures how much an externally bound object deforms a membrane around it (it is the inverse of the induced radius of curvature). To understand how and to what extent proteins or nanoparticles induce a local curvature, aaMD is an effective and unique approach. Pezeshkian et al. used aaMD to investigate the curvature-inducing capacity of the B subunit of Shiga (STxB) and cholera toxin (CTxB) [21,27]. They found that STxB induces a curvature of 0.07 nm^−1^ (corresponding to 0.035 nm^−1^ mean curvature), while for CTxB, this value is 0.05 nm^−1^ (Figure 1A and Figure 2B). It was also found that despite the different molecular designs of STxB and CTxB, the structure of a complex containing each toxin and its glycosphingolipid receptors, i.e., CTxB-5GM1 and STxB-15Gb3, induces a local curvature based on a similar principle. The combination of the positioning of specific binding units and protein shape provides a structural motif for curvature generation, which also has been termed the glycolipid–lectin (GL-Lect) hypothesis [10,11]. This new mechanism can be exploited for the design of drug delivery vehicles [21,27]. Interestingly, a recent experimental investigation using super-resolution and polarized localization microscopy on different CTxB mutants, each capable of binding to a different number of GM1, supports this picture and provides strong evidence that the CTxB-GM1 complex is the driver of the membrane curvature, not the CTxB alone [39,40]. It is worth noticing that both CTxB and STxB can enter a targeted cell by inducing long and narrow tubular membrane invaginations [5,41]. An important lesson from these studies is that the capacity of bacterial toxins to induce local membrane curvature within a specific range is crucial for their entry, and therefore provides a tuning parameter for the rational design of drug delivery vehicles. Curvature generation is also used to drive the endocytosis of endogenous proteins via the GL-Lect mechanism (Refs. [42,43,44], reviewed in [10]).

Overall, aaMD is a highly efficient and robust method to investigate toxin proteins and their interactions with the host membrane with an atomistic level of resolution. Nevertheless, there is a major limitation with aaMD: it is only applicable for dynamics at short times and on small length scales and is therefore only applicable to study single bacterial toxin particles and their interaction with a host membrane.

## 4. Coarse-Grain Simulations 

To capture biomolecular dynamics and interactions beyond the scales reachable with aaMD, coarse-grained (CG) models are at play. In this approach, a group of atoms is represented by an effective interaction site that results in a reduced system size (fewer number of particles), the removal of the fast degrees of freedom (allowing the use of larger time steps in the integration of the equations of motion), and smoother energy landscapes (faster dynamics) [45,46]. Owing to these inherent characteristics, CG models have become a powerful computational technique for biomolecular investigations in the length and time scales up to 100s of nm and 100s of microsecond, respectively. 

In the domain of toxin protein entry, these models are well suited to investigate the interactions between several toxic proteins, especially when they are bound to a biomembrane. For instance, the mechanism of STxB clustering on biomembranes has been explored using the dissipative particle dynamics method (a type of CG modeling) [22,47]. In this study, the authors ignore all the molecular details of STxB and represent it as a 7 nm diameter rigid pentagonal nanoparticle with an affinity to strongly bind to the biomembrane surface, a generic minimal model for rigid peripheral membrane proteins (Figure 1B). This simple model finds that the tight binding of sufficiently large nanoparticles is enough for them to cluster on the membrane surface, even in the absence of any direct interaction. The origin of the clustering mechanism is the ability of the nanoparticle to locally suppress membrane shape fluctuations; in response, the membrane generates an attractive force among these particles that drive them to cluster to increase the conformational entropy of the membrane, a mechanism which is termed the thermal Casimir-like force [48]. Interestingly, it is found experimentally that STxB clustering on membranes is strongly correlated to toxin tight binding to the membrane surface, and this clustering is necessary for the formation of tubular membrane invaginations, which is the initial step of STxB entry [22]. In another study, Reynwar et al. [49] used a solvent-free CG model [50] and showed that nanoparticles could aggregate (curvature mediated forces) and drive the membrane to vesiculate (a model for endocytic pit formation) when they induce large local membrane curvature. The importance of these results and methods is that they are generic and do not depend on a specific atomistic structure, and therefore provide design principles to learn from toxin proteins to engineer targeted drug delivery vehicles that can cross the cellular membrane effectively.

Another class of coarse-grained models aims to keep a certain degree of chemical specificity and atomistic accuracy while still benefiting from the speedups of CG modeling [51]. The forcefield of these methods is continually being optimized to make better predictions of realistic systems [52]. The Martini model, a popular CG model that maintains chemical specificity [53] (Figure 1B), has been used to investigate the effect of CTxB and its cellular receptor, i.e., GM1 on lipid bilayer phases and structures [54,55], and the binding of STxB and lectin I from Pseudomonas aeruginosa to membranes containing Gb3 [56]. Flores-Canales et al. [57] have created a CG model based on atomistic structure, to study the interactions of the diphtheria toxin translocation domain with biomembranes. 

Overall, CG models are versatile tools to investigate the mechanism of bacterial toxin entry into the host cell. On the one hand, they can be used as a simplified model to obtain generic physical features for the design of synthetic drug delivery vehicles; on the other hand, they can be exploited to investigate toxic-related biological mechanisms at near atomistic resolution. 

## 5. Mesoscopic Models

Many toxin proteins create their endocytic pit through cooperative actions of the cargo initiated by binding to its cellular receptor on the targeted cell membrane [38]. For instance, it is well established that via the GL-Lect mechanism, pathogenic lectins, e.g., Shiga and cholera toxins, generate tubular membrane invaginations by binding to their cellular glycolipid receptors and subsequently inducing local membrane curvature and clustering [5,41]. The GL-Lect mechanism has also been proposed to be important for cellular lectins [58]. To computationally investigate such processes, aaMD and CG models are expensive and inefficient. Instead, macroscopic modes in which the molecular details are ignored altogether and the membrane is typically represented by a continuous surface and the proteins, as one or few interacting particles, are better suited and a pragmatic choice [20,59,60,61]. A mesoscopic model based on dynamically triangulated surfaces (DTSs) has been used to investigate tubular membrane invagination induced by STxB [21]. In this study, the authors modeled each STxB protein by one particle and obtained the model parameters from simulations at higher resolution scales through a consecutive multiscale simulation scheme (Figure 1C). Interestingly, they not only reproduced the experimentally observed tubular membrane invaginations, but they also linked the molecular features of STxB particles to characteristic shapes of the invaginated membrane, which is indeed inaccessible to current experimental techniques. 

Another interesting physical phenomenon that is relevant to the generation of the endocytic pit, is membrane softening induced by membrane proteins. A membrane containing curvature-inducing proteins that are weakly interacting is effectively softer and above a certain concentration threshold, the bending rigidity diminishes. Therefore, the membrane no longer resists any applied force (curvature instability point). Mesoscopic simulations show that indeed, curvature-inducing proteins soften membranes; however, even below an instability concentration, membrane vesiculation (endocytic-like structures) and protein segregation takes place [62]. This mechanism is a result of cooperative action of curvature-active proteins on an undulating membrane and, therefore, cannot be captured in systems that only contain a few proteins, which is the limit of aaMD and CG simulations. 

Overall, mesoscopic modeling is a necessary type of simulation that fills a large gap in biosimulation methods. Additionally, through a multiscale framework, they can link molecular features to experimentally observable macroscopic properties, allowing for the better and more accurate interpretation of experimental findings. One of the notorious features of mesoscopic approaches is that they cannot provide any information about local properties at the single protein level. However, using back mapping software, e.g., TS2CG [63], their structure can be transferred to higher resolution models (Figure 1D).

## 6. Summary and Outlook

Computer modeling techniques have become an indispensable investigatory tool for exploring biological processes on a wide range of time and length scales. Owing to the rapid technological progress in hardware and software, and efficient algorithms and force fields, they have emerged as a new field, filling the gap between the theoretical and experimental investigations. Indeed, the mechanism of action of bacterial toxins is a good example. In particular, with the development of new multiscale simulation approaches, valuable information about their mechanism of action and entry pathway has been obtained. Multiscale simulations provide a scheme for analyzing the cascade of successive complex processes of toxin action from the molecular details to the effect of membrane structural deformations and fluctuations. With the fast progress in the multiscale simulation techniques, these methods will become a standard tool complementary to experiments in the research of toxins. Additionally, simulations can provide certain collective properties emerging from molecular features that can be used as tuning parameters in the rational design and engineering of drug delivery vehicles. For instance, the capacity of these vehicles to induce a certain range of curvature and to harness the membrane fluctuations is an important feature for endocytosis and indeed can be used to build, for instance, membrane binding nanoparticles with a prescribed curvature using DNA origami.

A major limitation of many computational results is oversimplification and idealization. So far, computational studies have considered toxin particles interacting with simple bilayers that are isolated entities, whereas, in realistic systems, the plasma membrane has a highly complex composition, is tethered to cortical actin, and is in direct contact with the cytoplasm that constitutes a highly crowded and heterogeneous environment. Therefore, the future challenge will be to increase the complexity of membrane composition and add relevant influences from the surrounding environment. This will bring simulated systems closer to biological reality and greatly accelerate their usefulness.

## Figures and Tables

**Figure 1 toxins-13-00449-f001:**
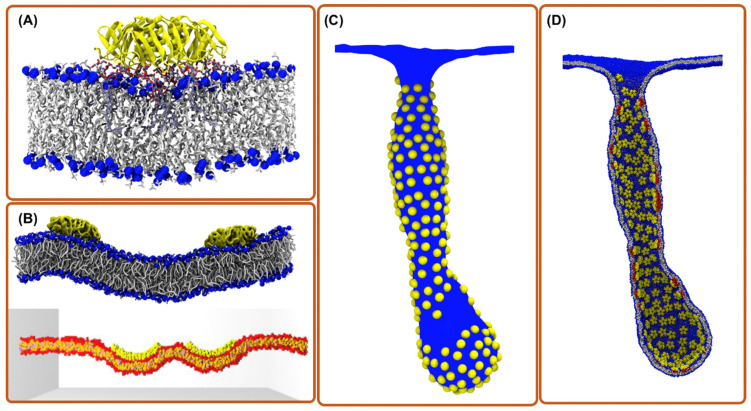
Computational simulation methods for exploring toxin entry into the host cell. (**A**) aaMD simulation of a single Shiga toxin bound to the POPC bilayer containing a fraction of Gb3 lipids. The binding of the toxin to Gb3 generates a small local membrane curvature and suppresses membrane fluctuations [21]. (**B**) CG simulations of two toxin particles bound to a membrane; (top) Martini model where chemical moieties are retained; (bottom) DPD model where a still simpler forcefield is employed [22]. (**C**) Mesoscopic simulations of hundreds of toxin particles remodeling the membrane shape (picture is obtained from ref. [13]). (**D**) Back mapping structure from “C” to CG model for investigations with molecular detail (picture is obtained from ref. [13]).

**Figure 2 toxins-13-00449-f002:**
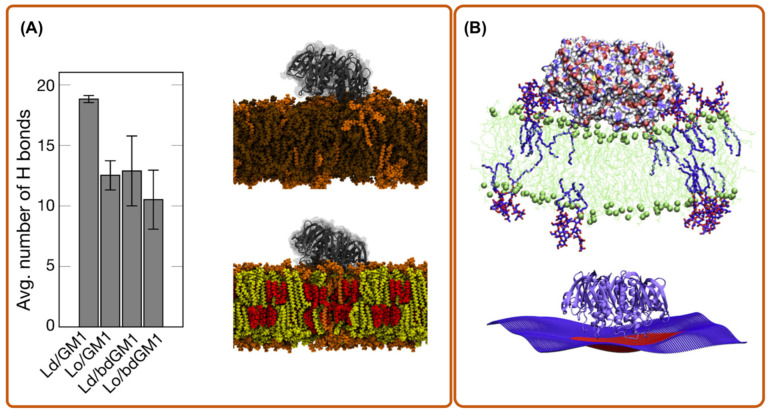
aaMD simulation of a cholera toxin bound to a bilayer. (**A**) the dependence of GM1-CTxB binding on membrane environment in simulations of systems with CTxB (picture is made from ref. [34]); (**B**) CTxB induce local membrane curvature upon binding to a lipid bilayer containing a fraction of GM1 (picture is made from ref. [27]). Lo and Ld refers to liquid order and disorder phases, respectively.

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
