# Peer review of "Computational Approaches to Explore Bacterial Toxin Entry into the Host Cell"

_toxins, 2021, doi:10.3390/toxins13070449_

Round 1

Reviewer 1 Report

This paper describes the computational approaches to explore bacterial toxin entry into the cell. First, the paper highlights the strength of computational methods for exploring different stages of the toxin protein entry mechanism. Next, the author summarizes recent developments that significantly advance our understanding of glcycolipid-lectin endocytosis of bacterial Shiga and cholera toxins. The method is also applicable to the design of membrane-penetrating nanoparticles and the phenomenon of proteins phase separation at the surface of the membrane.

I am a biologist studying bacterial toxin entry. While I am not an expert on the computational method, I expect the computational approaches to study the entry mechanism. The entry mechanism is classified into several steps: (1) Receptor binding, (2) Membrane attach (3) Membrane pore formation (4) Translocation of the substrate toxin. Each step needs to study more. Otherwise, the computational methods can not work well. Moreover, the present computational method can not calculate in a long time. Thus it restricts the effectiveness of the method.

The author tried to review the computational approaches for the cell entry of cholera toxin and shiga toxin: all atom molecular dynamics, coarse grain simulation and mesoscopic method.

Unfortunately, while I am interested in the computational approaches, this review did not provide that the computational method can cover our experimental questions or raise the novel principle of toxin entry.

Furthermore, it is not easy to understand the interests of the study from the sentence and figure.

page 3

in the range of 10s of nm and microsecond timescale>I am not sure the meaning of 10s of nm.

Please explain this.

Figure 1

The computational method can provide the dynamic motion of the biomolecular complex. However, in this meaning, the figure can not provide the effectiveness of the dynamic motion.  

Author Response

We would like to thank the reviewer for her/his thorough reading of our manuscript and detailed criticism. In the following, we address the specific points raised by each reviewer and hope that our answers are fulfilling. Our responses are given in bold.

I am a biologist studying bacterial toxin entry. While I am not an expert on the computational method, I expect the computational approaches to study the entry mechanism. The entry mechanism is classified into several steps: (1) Receptor binding, (2) Membrane attach (3) Membrane pore formation (4) Translocation of the substrate toxin. Each step needs to study more. Otherwise, the computational methods can not work well. Moreover, the present computational method can not calculate in a long time. Thus it restricts the effectiveness of the method.

We believe that the points raised by referee have been addressed well in the previous manuscript. However, a few important points appear to have been missed by the referee. First of all, although aaMD and cgMD are limited in time and length scale, the mesoscopic models are very close to experimental system size and time scales. Moreover, in several sections we have made it clear that for exploring toxin entry, several of these methods are needed and they can be linked through effective multiscale schemes. Additionally, although some bacterial toxins attach to the targeted cell by inducing membrane pore, majority of them internalize through endocytosis which has been the focus of this review. We believe that all other steps mentioned by the referee are described well in the previous manuscript. However, we note that the number of experimental works focused on bacterial toxins is by far larger than the ones done using computational methods and therefore we are limited to report the works that are already done, not to provide new results as this is a review paper. With this publication we hope to attract more experimentalist who focus on toxin systems to collaborate with computational groups. The simple language used in this article is for this reason.

The author tried to review the computational approaches for the cell entry of cholera toxin and shiga toxin: all atom molecular dynamics, coarse grain simulation and mesoscopic method. Unfortunately, while I am interested in the computational approaches, this review did not provide that the computational method can cover our experimental questions or raise the novel principle of toxin entry. Furthermore, it is not easy to understand the interests of the study from the sentence and figure.

The reviewer did not mention what are “their” experimental questions that she/he expect to be answered by simulations. Therefore, we cannot further comment on that.

In the manuscript, we discuss several fundamental principles that are involved in toxin entry and were revealed by simulations. These works have been in close contact with several well-known cell biologists that are investigating bacterial toxin experimentally and they have indeed acknowledged that these results are novel and highly interesting.

page 3

in the range of 10s of nm and microsecond timescale>I am not sure the meaning of 10s of nm.

Please explain this.

10s is used to describe several multiples of 10, in this case a distance of the order of 20-50 nm.

Figure 1

The computational method can provide the dynamic motion of the biomolecular complex. However, in this meaning, the figure can not provide the effectiveness of the dynamic motion.  

We agree that simulations can provide dynamic motion of the biomolecular complex and we agree that an image (snapshot in this case) cannot provide that. Numerous experimentally

 measurable quantities characterizing the system dynamics can be estimated from simulations. However, making animations is only one of the benefit of using computational methods. 

Reviewer 2 Report

The manuscript toxins-1253855 is a review survey focused on bacterial toxins and their interaction with membranes, investigated by using computer simulations that include all-atom MD, coarse-grained simulations, and mesoscopic models. It is a nice and sweet piece of work, rather clear in its scope and easy to read, with a number of exceptions highlighted below. In the following, I will provide a minor suggestions that can be optionally be considered to improve the manuscript.

- General remark: the three type of simulation techniques used (aaMD, CG, and mesoscopic simulations) might already be mentioned in the Abstract. This will make clearer to the reader what to expect about this review.

- Line 17, “the phenomenon of protein phase separation”. It would be clearer “the study of the phenomenon...”.

- Line 38, “vesicle techniques”. This is puzzling to me.

- Line 56, “(see below)”. It is not clear what this refers to; I would suggest to either expand it a little bit or delete it.

- Line 60-62, “This suggest that generic physical properties of the membrane are exploited, where computational modelling is method of choice for such their analysis”. This whole part is garbled.

- Lines 63, “for exploring different stages of the bacterial toxin entry mechanism”. The hint to “mechanism stages” is a bit unclear.

- Line 70, “With the current state of human knowledge, ...”. It is obscure to me why this incipit is deemed necessary, and what is supposed to be its purpose.

- Line 85, “while they may fail completely outside these ranges”. The techniques mentioned do not even fail; rather, the concept is that they are inapplicable outside the ranges described.

- Line 129, “certain degrees of chemical specificity are preserved”. This is too vague.

- Line 130, “only non-specific binding is included”. Again, too vague.

- Line 144, “membrane bud radius”. I do not know what a “bud radius” is.

- Line 162, “LoMemCI can be used as a tuning parameter”. I have no idea what concept was meant to be expressed here.

- Line 175, “allowing use of larger time steps”. Maybe adding “in the integration of the equations of motion” would be better.

- Line 204, “These methods are becoming more optimized”. More optimized in what?!

- Line 211-212, “On one end, ...on the other end”. Use “hand”.

- Line 238 “curvature instability point”. Why is this bold in the text?

- Line 246, “biosimuation”. Typo.

- Line 268, “can be used to design, for instance, in DNA nanotechnology”. Too vague.

- References: the mixed use of lowercase and uppercase letters in the titles should be checked. Also, I never saw before a reference reported as “accepted in principle” (line 299).

Author Response

We would like to thank the reviewer for her/his thorough reading of our manuscript and detailed criticism. We have changed the manuscript according to the reviewers’ comments. In the following, we address the specific points raised by each reviewer and hope that our answers are fulfilling. Our responses are given in bold.

Reviewer 2:

Comments and Suggestions for Authors

The manuscript toxins-1253855 is a review survey focused on bacterial toxins and their interaction with membranes, investigated by using computer simulations that include all-atom MD, coarse-grained simulations, and mesoscopic models. It is a nice and sweet piece of work, rather clear in its scope and easy to read, with a number of exceptions highlighted below. In the following, I will provide a minor suggestions that can be optionally be considered to improve the manuscript.

We thank the reviewer for her/his positive opinion on our work. We have changed the manuscript according to the reviewer comments and hope it is satisfactory. 

- General remark: the three type of simulation techniques used (aaMD, CG, and mesoscopic simulations) might already be mentioned in the Abstract. This will make clearer to the reader what to expect about this review.

As the reviewer requested this has been added to the abstract.

- Line 17, “the phenomenon of protein phase separation”. It would be clearer “the study of the phenomenon...”.

 It has been changed accordingly

- Line 38, “vesicle techniques”. This is puzzling to me.

 We thank the referee for mentioning this. We have removed these words from the manuscript since they refer to experimental setups involving GUVs rather than a technique.     

- Line 56, “(see below)”. It is not clear what this refers to; I would suggest to either expand it a little bit or delete it.

We agree. It is an unnecessary remark, so we have deleted it.

- Line 60-62, “This suggest that generic physical properties of the membrane are exploited, where computational modelling is method of choice for such their analysis”. This whole part is garbled.

 We have modified this so that it now reads: “This suggests that generic physical properties of the membrane are exploited and computer simulations are an excellent tool for uncovering them.”

- Lines 63, “for exploring different stages of the bacterial toxin entry mechanism”. The hint to “mechanism stages” is a bit unclear.

We have expanded on 'mechanism stages' in the revised manuscript.

- Line 70, “With the current state of human knowledge, ...”. It is obscure to me why this incipit is deemed necessary, and what is supposed to be its purpose.

 We agree and the whole sentence in the manuscript has been changed to: “Despite the colossal complexity of biological systems, they are still subject to the laws of physics”.

- Line 85, “while they may fail completely outside these ranges”. The techniques mentioned do not even fail; rather, the concept is that they are inapplicable outside the ranges described.

The reviewer is correct. In some cases, they are inapplicable outside these ranges in other cases we do not have the computational power to analyze the models outside the specified ranges. We have changed it to: “while they are inapplicable outside these ranges”

- Line 129, “certain degrees of chemical specificity are preserved”. This is too vague.

 It is now changed to “Martini model where chemical moieties are retained

- Line 130, “only non-specific binding is included”. Again, too vague.

It is changed to “a still simpler forcefield is employed”

- Line 144, “membrane bud radius”. I do not know what a “bud radius” is.

This has been clarified.

- Line 162, “LoMemCI can be used as a tuning parameter”. I have no idea what concept was meant to be expressed here.

 This part has been changed and clarified. Also, we have replaced all ocurrences of “LoMemCI” with local membrane curvature imprint.

- Line 175, “allowing use of larger time steps”. Maybe adding “in the integration of the equations of motion” would be better.

 This has been added.

- Line 204, “These methods are becoming more optimized”. More optimized in what?!

 We changed the sentence to “The forcefield of these methods are continually being optimized to make better predictions of realistic systems”

- Line 211-212, “On one end, ...on the other end”. Use “hand”.

We thank the reviewer for pointing this out and have changed it accordingly.

- Line 238 “curvature instability point”. Why is this bold in the text?

This is fixed now.

- Line 246, “biosimuation”. Typo.

 We thank the referee for noticing the error. This typo is corrected now.

- Line 268, “can be used to design, for instance, in DNA nanotechnology”. Too vague.

We have clarified this in the manuscript and now reads as:

“For instance, the capacity of these vehicles to induce a certain range of curvature and to harness the membrane fluctuations is an important feature for endocytosis and indeed can be used to build, for instance, membrane binding nanoparticles with a prescribed curvature using DNA origami”.

- References: the mixed use of lowercase and uppercase letters in the titles should be checked. Also, I never saw before a reference reported as “accepted in principle” (line 299).

The references are made using Endnote software. The Journal will change this when they use the journal style.

Regarding lin e299, we expected that the mentioned article would be accepted by the time this article is reviewed and we would then add the full reference. We have now done this.

Reviewer 3 Report

In this literature review, the author(s) addresses the exploration of bacterial toxin entry into the host cells by employing computational approaches. While such modern approaches are essential for overcoming experimental limitations, they also possess predictive capabilities. Consequently, their application ranges from fundamental science to biomedical applications and nanotechnologies, which makes the content palatable for broad audiences.

The justification for using computational methods and their strength for a better understanding of the toxin/host interactions are succinctly described in section 1 ( Introduction) and 2 (Computational methods). Recent advances in this field made possible by specific simulations and models are further described and summarized in the dedicated sections 3-5.

In my opinion, this review provides the reader with a good, comprehensive overview of this important topic. Significant areas of recent research are surveyed and sufficiently highlighted, and future improvements needed to overcome the current technical limitations are presented.

I think this review may be improved by addition to each of the simulation/model sections of a few more specific details of one or more the selected topics, accompanied by figures from the referenced work. The current manuscript provides a good description of prior achievements, and the interpretation of computational results is sufficiently presented in the main text. However, the visual impact to readers may be improved by addition of a few figures related to the described context. For example, the aaMD section may include such figures related to computational explorations on CTxB binding, or molecular probes from the referenced articles. The only figure presented in the manuscript, although relevant for the presented material, is also from a review article.

Therefore, I consider that this review is suitable for publication in Toxins. I would really appreciate if the author(s) will address my minor concerns and revise the manuscript accordingly.

Author Response

We would like to thank the reviewers for her/his thorough reading of our manuscript and detailed criticism. We have changed the manuscript according to the reviewers’ comments. In the following, we address the specific points raised by each reviewer and hope that our answers are fulfilling. Our responses are given in bold.

Reviewer3

Comments and Suggestions for Authors

In this literature review, the author(s) addresses the exploration of bacterial toxin entry into the host cells by employing computational approaches. While such modern approaches are essential for overcoming experimental limitations, they also possess predictive capabilities. Consequently, their application ranges from fundamental science to biomedical applications and nanotechnologies, which makes the content palatable for broad audiences.

The justification for using computational methods and their strength for a better understanding of the toxin/host interactions are succinctly described in section 1 ( Introduction) and 2 (Computational methods). Recent advances in this field made possible by specific simulations and models are further described and summarized in the dedicated sections 3-5.

In my opinion, this review provides the reader with a good, comprehensive overview of this important topic. Significant areas of recent research are surveyed and sufficiently highlighted, and future improvements needed to overcome the current technical limitations are presented.

I think this review may be improved by addition to each of the simulation/model sections of a few more specific details of one or more the selected topics, accompanied by figures from the referenced work. The current manuscript provides a good description of prior achievements, and the interpretation of computational results is sufficiently presented in the main text. However, the visual impact to readers may be improved by addition of a few figures related to the described context. For example, the aaMD section may include such figures related to computational explorations on CTxB binding, or molecular probes from the referenced articles. The only figure presented in the manuscript, although relevant for the presented material, is also from a review article.

Therefore, I consider that this review is suitable for publication in Toxins. I would really appreciate if the author(s) will address my minor concerns and revise the manuscript accordingly.

We thank the reviewer for her/his positive opinion on our paper. We have added another figure (Figure 1) to the manuscript.

Round 2

Reviewer 1 Report

Unfortunately, I could find any significance, which we want to use to reveal the bacterial toxin entry mechanism.